# The footprint of campaign strategies in Farsi Twitter: A case for 2021 Iranian presidential election

**Saeedeh Mohammadi[1,2], Parham Moradi[2], S. Mahdi Firouzabadi[3], Gholamreza Jafari[1,4]\***

**1** Physics Department, Shahid Beheshti University, Tehran, Iran, **2** Center for Complex Networks and Social Data Science, Tehran, Iran, **3** Institute for Cognitive and Brain Sciences, Shahid Beheshti University, Tehran, Iran, **4** Institute of Information Technology and Data Science, Irkutsk National Research Technical University, Irkutsk, Russia

\* g_jafari@sbu.ac.ir

**Data Availability Statement:** The data for this study was gathered through the Twitter Standard API and according to the Twitters Developer Agreement and Policy. The analysis is done on a macroscopic layer and no conclusion is made

## Abstract

The rise of social media accompanied by the Covid-19 Pandemic has instigated a shift in paradigm in the presidential campaigns in Iran from the real world to social media. Unlike previous presidential elections, there was a decrease in physical events and advertisements for the candidates; in turn, the online presence of presidential candidates is significantly increased. Farsi Twitter played a specific role in this matter, as it became the platform for creating political content. In this study, we found traces of organizational activities in Farsi Twitter, and our investigations reveal that the discussion network of the 2021 election is heterogeneous and highly polarized. However, unlike many other documented election cases in Iran and around the globe, communities of candidates' supporters are very close in one pole, and the other pole is for "Anti-voters" who endorse boycotting the election. With almost no reciprocal ties, these two poles form two echo chambers, one favoring the election and the other for voter suppression. Furthermore, a high presence of bot activity is observed among the most influential users in all of the involved communities.

## Introduction

The increasing number of online users who spend more and more time on social media has made these platforms a significant source of influence in modern societies. Initially, social media was seen as a way to amplify political and cultural grievances throughout the world, its role in organizing protests and giving a voice to the opposition in instances like the Arab Spring [1], the Green movement [2], the "Me Too" movement [3] and the Black Lives Matter movement [4], portrayed social media as a "liberation technology" [5]. The transparency provided by social media created a sense of trust in users to use these platforms for political means [6]. Later on, many studies discovered traces of online manipulation in socio-political events. Researchers suggested that political actors have invested in and used social media platforms to propagate their agenda and manipulate the public in their favor [7–9]. Bessi et al. [10] discovered such activities in the 2016 U.S. presidential election by identifying over 400000 bot

about a named or unnamed individual. The stored data is encrypted and information like username, user id, and screen name are removed. The graphs are available in https://ccnsd.ir/twitter-project/.

**Funding:** The author(s) received no specific funding for this work.

**Competing interests:** The authors have declared that no competing interests exist.

accounts on Twitter that participated in the election discussion. Similarly, Bastos et al. [11] focused on the 13493 bot-like Twitter accounts that tweeted during the UK European Union membership referendum debate to identify online manipulation during Brexit. Other studies have also focused on identifying automated accounts to discover coordinated actions during socio-political events [12–14]. However, bot detection requires sophisticated machine learning algorithms, and comprehensive statistical and text analysis while still in some cases have not been successful [15–19]. Network science has been used along with other statistical approaches to identify the spread of fake news [20], detect voter suppression [21], and echo chambers [22]. In this article, we show that by merely observing the evolution of communities in the retweet network, it is possible to not only detect some campaign's tactics but also get an overall accurate representation of the prominent political players who competed to gain voter's attention.

The political landscape of Farsi Twitter during Iranian elections in 2013 and 2017 has been investigated by Khazraee [23] and Kermani et al. [24] Both these studies have focused on the genuine users' activities and how voters expressed themselves on Twitter during this time. The increase in the online activities of the conservative party is observed in the 2017 election, however, in both cases, the majority of users are in favor of the reformist party. However, the significant presence and participation of conservative supporters are observed during this election. The majority of users who were involved in the 2021 election discussion have been identified as conservative supporters. These studies gave an overall picture of how Farsi Twitter users interacted and expressed themselves during elections, we attempt to add to these researches by examining the actions of political groups and detecting the presence of online manipulation within this discussion. The presence of political campaigns during elections on Twitter has been heavily studied [25–27]. Election campaigns use different methods like employing fake users, trolls, bots, and sophisticated digital marketing techniques for their operations during elections [10, 12]. Detection of these organized activities and inauthentic users is significant in protecting the integrity of democracies. Farzam et al. [28] used network analysis to identify online manipulation in different types of discussions on Farsi Twitter. There is a lack of research when it comes to organized activity and user manipulation through Farsi Twitter in a consequential era like a presidential election.

In this article, we examine the political debate in Farsi Twitter in the heat of the 2021 Iran presidential election. It is important to mention, the 2021 Iranian presidential election was between the following candidates from the conservative party; Ebrahim Raisi (president-elect), Saeed Jalili (withdrew in favor of Raisi), Alireza Zakani (withdrew in favor of Raisi), Mohsen Rezaii and Amir Hossein Ghazizadeh, and the following candidates from the reformist party; Abdolnaser Hemmati and Mohsen Mehralizadeh (withdrew in favor of Hemmati). It is important to note that the candidacy of many politicians was rejected most of whom were from the reformist party which resulted in certain groups boycotting the election. The following politicians were rejected a presidential candidacy and were outspoken during the election. Mahmud Ahmadinejad and Saeed Jalili. We use the trending hashtags regarding the election from seven-week before to one week after the election. While the opposition discourse was still active in the Twitter election debate, we observed a significant rise in tweets favoring the conservative candidates, particularly several days before the election. The retweet networks about this discussion are highly polarized, however, the presidential candidates are together on one side and the other pole consists of the opposition. This could point to the political structure in the real Iranian political space [29].

We present evidence based on network analysis, the presence of high levels of user manipulation through hashtags led by inorganic users favoring almost all campaigns involved in the election. Our results imply that the trending hashtags of Farsi Twitter in such events merely represent public opinion since users who tweet hashtags generally have a political agenda. Our

study approves that analyzing network communities and their dynamics offers powerful means to investigate campaign strategies of different political groups.

## Materials and methods

In this section, we describe our approach to data gathering, then elaborate on the configuration of the networks, and finally explain our method to analyze the networks based on network topology.

### Data collection

We monitored trending hashtags daily and gathered 97 hashtags related to the 2021 Iran presidential election from the 29th of April to the 24th of June (7 weeks before the election to one week after). These trending hashtags represented topics related to the election. Most Hashtags were either sign of support or opposition to a candidate or the whole election process.

We constructed a "Twitter Machine" program that takes a keyword and an initial date as the input and stores any tweet, including that word, since the given date in an SQLite database. The tweets are made available through Twitter's Standard Search API. Twitter Machine deconstructs the JSON that API returns, records the necessary data in an SQLite database, and updates the user index in the PostgreSQL database. PostgreSQL is used as a relational database and contains every user participating in a conversation gathered by Twitter Machine. The users' information is regularly updated when Twitter Machine does a new task.

The Twitter Machine collected 8818675 tweets that included the hashtags relating to the election. 153115 users tweeted these posts over the mentioned eight weeks used in this study.

To analyze the transformation of user behavior over time, these data were divided into eight different periods (see Table 1). For each time frame, the number of tweets and users collected by Twitter Machine is listed.

To identify user authenticity, we used Botometer [30] and gathered Botscore, Complete Automation Probability (Cap), and Fake Followers for users. To improve the accuracy of the analysis, we used crowdsourcing to annotate over 1000 accounts manually. As a result, we noticed that users with CAP scores less than 0.01 are likely to be genuine users, and the ones with CAP scores higher than 0.75 are more likely automated or semi-automated accounts [28].

### Networks

Network analysis can be applied to understand whether the propagation of information regarding the election and user behavior are well represented in the election Hashtags. While the friendship network embodies valuable information about the users' social interactions, the

**Table 1. The different time frames of data collection are listed with the amounts of users and tweets collected during this time.** BPE and APE stand for before and after presidential election respectively.

| Label | Time Frame | No. tweets | No. users |
|:---:|:---|---:|---:|
| a | 7–5 Weeks BPE | 444,204 | 31,015 |
| b | 5–4 Weeks BPE | 893,868 | 47,316 |
| c | 4–3 Weeks BPE | 1,114,342 | 50,194 |
| d | 3–2 Weeks BPE | 1,591,757 | 65,325 |
| e | 2–1 Weeks BPE | 1,117,377 | 72,607 |
| f | 1 Week-2 Day BPE | 1,722,812 | 74,464 |
| g | 2 Day BPE-1 Day APE | 987,984 | 54,054 |
| h | 1 Day-1 Week APE | 478,363 | 49,549 |

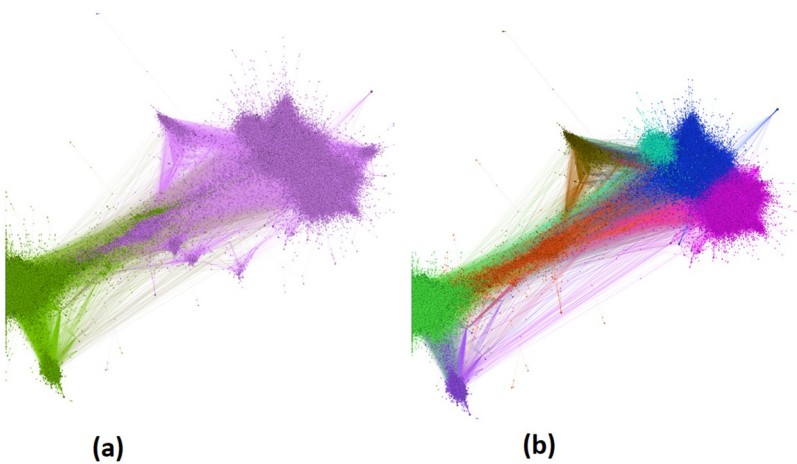

**(a)** **(b)**

**Fig 1. Retweet network of the election in Farsi twitter with different resolutions when calculating the modularity.** The colors represent the node's community that is detected by modularity. The size of the nodes is proportional to the accumulated number of retweets of the user. (a) resolution is set to 25 (b) resolution is set to 1.

retweet network seems to be better for this study. It would reveal information about the engagement of users surrounding specific events of the election. To ensure that we would only include politically engaged users, each node in the retweet network is a user that tweeted a post that included a hashtag relevant to the election. If user A retweets a post from user B at any point, an edge links user B to user A in the retweet network. In Fig 1, the retweet network for the whole duration of data collection is illustrated using the forceAtlas2 algorithm [31], where the size of each node is proportional to the user's accumulated retweet count. Note that this number differs from the node's in-degree since the network is only representative of the specific time period and the retweet count of the user's tweets represents the number of retweets overall.

The node's color in the network indicates to which community a node belongs. The community detection is done using the greedy modularity algorithms [32].

The resolution set in the modularity [33] specifies the size of the communities. The higher the resolution, the communities get more extensive and include more nodes. Eventually, the number of communities becomes smaller. Fig 1(a) illustrates the retweet network with a resolution set to 25 that has detected the two major communities, whereas, in Fig 1(b) the communities are more but smaller with the resolution set to 1.

Some of the network's essential structural characteristics are measured to study and compare the structure of different communities. These properties include; Average clustering coefficient [34], which gives us an insight into the linking structure within a community; modularity [32], which measures the presence of smaller clusters within a community; the average shortest path, that would indicate the small worldness of a community and assortativity [35], measures each node's likelihood to link with similar nodes, for instance, the probability of a hub linking to another hub node. These properties would provide a vivid representation of the structural differences and similarities of different communities.

Many studies indicate a dual nature for Twitter. One as a medium for social communication, exchange of opinions with more reciprocal ties, and echo-chamber-like spaces [36, 37]. The other is a medium to receive information and a news platform with more one-sided relations [38, 39]. To investigate this idea, we compare the reciprocal and non-reciprocal retweet

networks of the election. The reciprocal network is a subgraph of the non-reciprocal network. Each edge in this network is among users whom at least retweeted each other once.

As Colleoni outlines, the network based on reciprocal ties might indicate higher levels of homophily and an echo-chamber-like environment [40]. We use the external-internal (E-I) index to measure the isolation and embedding of communities in each network. The E-I index is the difference between the number of external ties and internal ties divided by the total number of ties. the index ranges from −1 (all ties are internal to the group) to +1 (all ties are external to the group) [41]. As we expected, our results indicate a higher level of homophily in reciprocal networks which means that the two-way communication about the election is highly confined within same-minded isolated communities.

In the following section, we will discuss the detailed properties of the two retweet networks.

## Results and discussion

In this section, the retweet network of the election is analyzed. A retweet network is considered to be the network of information dissemination.

To detect the supporters of each candidate, first, we use modularity classes to separate nodes into different clusters (For this article, only the seven largest communities are examined, and smaller communities are excluded.) Next, we manually investigated the tweets posted by the top 10 nodes with the most retweet from each community during the election period. After detecting these users' political preferences, we labelled each cluster accordingly. Labels were about promoting or advocating specific candidates or a political stance (Anti voting or Pro voting) are called their community. Afterwards, we investigate the accuracy of assigned labels in representing each community by taking random samples from each community. We use Cochran's formula [42] to calculate the required sample size of a finite sample with 90% accuracy. The average accuracy of labels in all communities of the retweet networks is above 80%. In addition, as we get closer to the election, the accuracy increases from 86% to 93.5%. This finding suggests that as we get closer to the election, the assigned label is more accurate in describing the political preference of network communities.

In Fig 2, the retweet networks are illustrated with ForceAtlas2 [31] as the layout algorithm. Each community is defined with different colors. Each network represents the flow of information about the election in a particular period, specified in the timeline on the left.

The anti-community (users that declare their refusal to vote and encourage others to boycott the election) is on one end of the discussion, and the opposite end, the pro community (users that encourage others to vote and claim they would as well), Jalili community, Mohammad community, and Raisi community. This state is valid throughout the timeline. While Mohammad and Jalili communities are not active in the days closer to the election and after, the Raisi community preserves their place opposite the anti-community.

It is evident in Fig 2 that the anti-community constitutes multiple clusters in most networks. The community detection shows 2 clusters in the anti-community, which we call; the main body and the tail. The hashtags widely used in both of these clusters are examined. The most used hashtags in the tail cluster are hashtags that are more likely used by MEK supporters (Mojahedin E Khalq or People's Mojahedin, a militia group located outside of Iran that advocates overthrowing the Iranian regime [43]), which would suggest that this cluster belongs to MEK supporters. However, since both these clusters propagate the same message, boycotting the election, we refer to both as the anti-community.

By the end of the vetting process, the guardian council published the list of approved candidates on the 25th of May. Many supporters were disappointed as their candidate could not participate in the election. Therefore, they expressed their frustration on Twitter. The

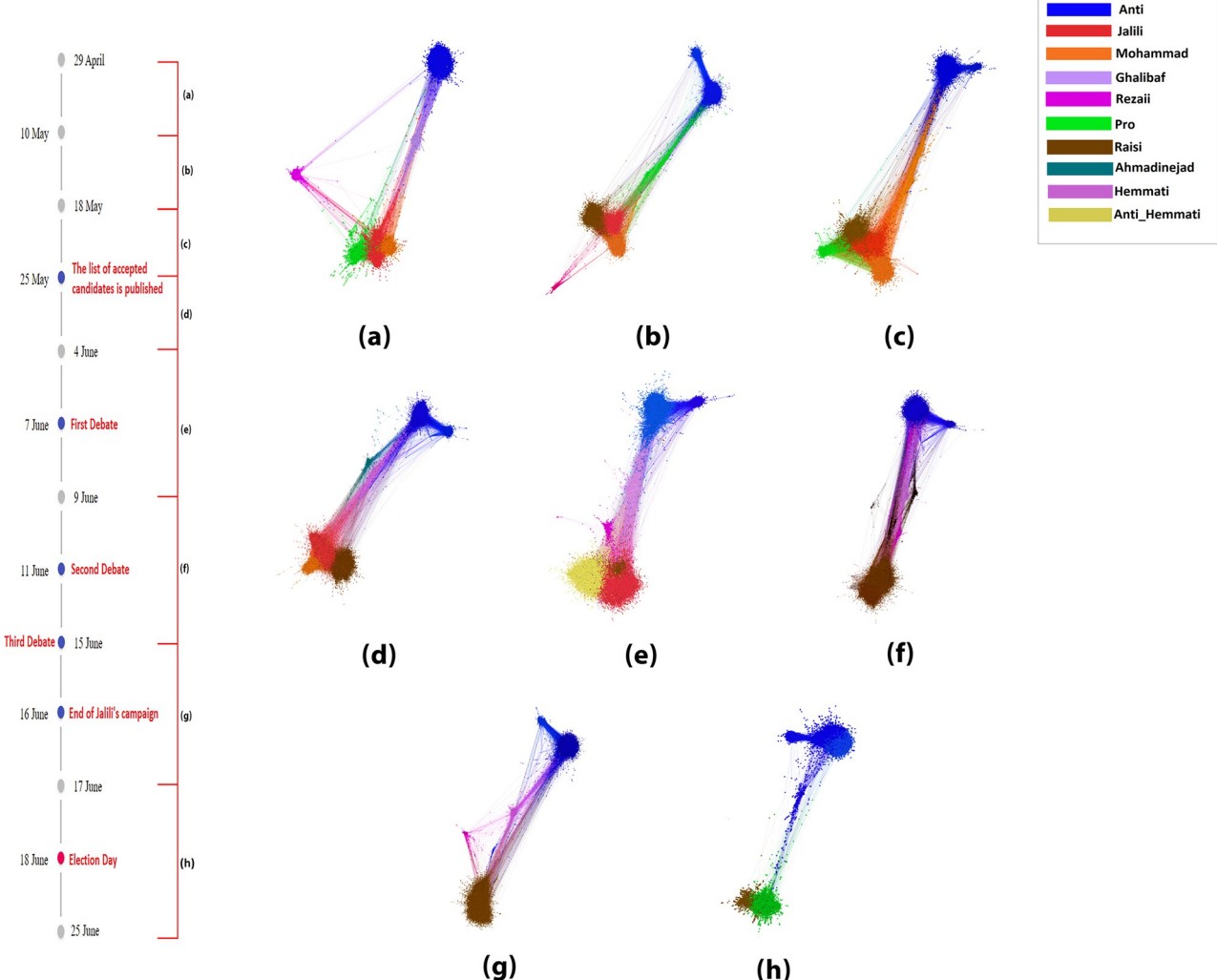

**Fig 2. An illustration of the seven biggest communities of retweet networks throughout data collection.** On the left, a timeline of dates and events surrounding the election is illustrated; The period of each retweet network is specified as well. Each node's color corresponds to the community to which it belongs.

Ahmadinejad community's presence in the retweet network (d), somewhere between the two poles and closer to the anti-community, points to the anger of his supporters about this decision. On the other hand, Mohammad's supporters, who have also declined a presidential candidacy, preserve their place opposite the anti-community and close to Jalili and Raisi's communities.

During the 2021 presidential election debates, many hashtags started trending in opposition to the Reformists candidate, Hemmati. The content of these hashtags did not show any support for any particular candidate and just criticised Hemmati. Hence, users who used these hashtags during the debates are labelled "anti-Hemmati," and their community is observed in the (d) network. It is important to note that this community is very close to Jalili and Raisi's community. Fig 4(Above) demonstrates which community these users belonged to in the previous network, which shows most of them belonged to the Raisi community. This could

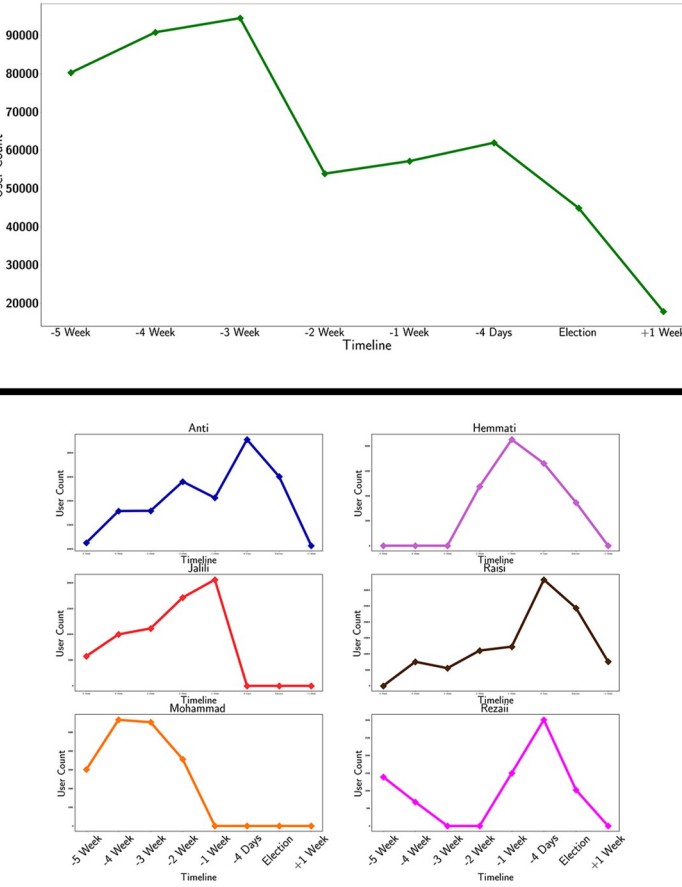

**Fig 3.** Top: The amount of users that participated in the retweet network about the election by tweeting a trending hashtag. Bottom: The amount of participating users separated by communities.

explain the reduction of Raisi's community in this network. We would merge "anti-Hemmati" and Raisi's communities for future investigations.

As it gets closer to the election itself, some candidates' support is reduced and increased for some others. For instance, Mohammad's community is not in the retweet network after the (d) network, or Rezaii's community disappears during (c) and (d) networks and reappears. It is important to note that the fact that these communities are not in the retweet networks in Fig 2 does not mean that they were no support for them on Twitter during that time; it just implies that the support for them was reduced to the point that they did not appear in the seven largest modularity class of the original retweet network. Fig 3 offers the number of users involved in the retweet network in Farsi Twitter during the data collection period. The user count of each community has been done for each period.

An interesting finding from Figs 2 and 3 is in the (f) network, one week to 4 days before the election, where Jalili's supporters decrease significantly and are omitted from major communities. This omission happens while Jalili ends his campaign two days later, suggesting that this sudden lack of support on social media compels Jalili to end his campaign, or the decision was made a week before the election as a systematical move from Jalili's team. Furthermore, there is a significant rise in Raisi's community as Jalili's declines, suggesting that the users who formerly belonged to Jalili's community have merged with Raisi's supporters. This claim is proved in Fig 4 (Below).

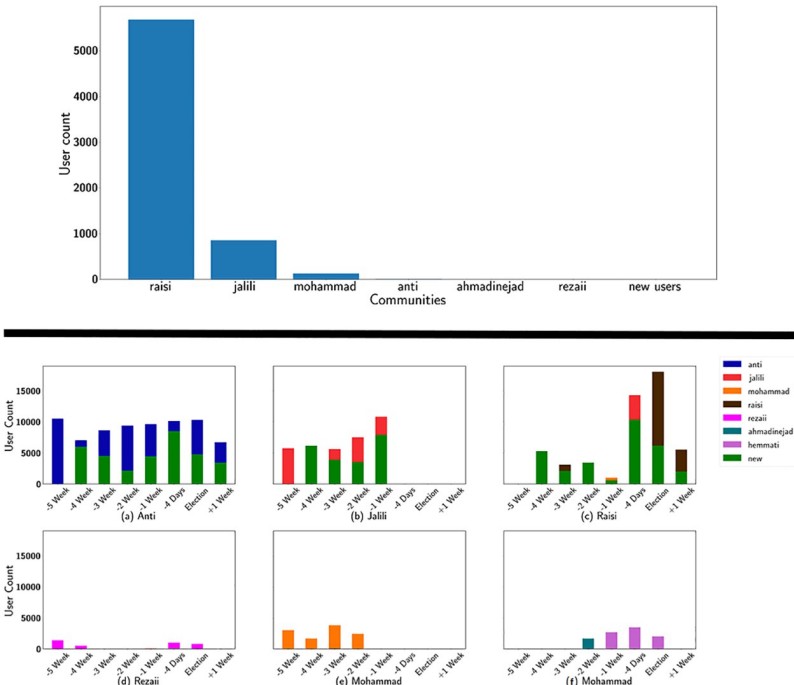

**Fig 4.** (Above) User distribution of the anti-hemmati community. (Below) The user distribution over the timeline in each community. The colors refer to which community the users belonged to in the previous tick. The communities are: (a) Anti, (b) Jalili, (c) Raisi, (d) Rezaii, (e) Mohammad, (f) Mohammad.

Fig 4 depicts the way communities have grown or contracted during this time frame. Each bar shows the number of users in a specific community, and the colors illustrate which community they belonged in the previous tick. New users who were not in the previous communities are also demonstrated. Fig 4(a) shows how Jalili's community merged into Raisi's community four days before the election; the same also happens as Mohammad's community merges into Raisi's community. Additionally, the lack of new users or users from other communities in Rezaii and Mohammad's community could point to organized activity.

Fig 1(a) illustrates two central communities of the whole network. The E-I index of this network is about -0.98, which indicates the network is very polarised and heterogeneous. Fig 5 illustrates the reciprocal subgraph of this network, which shows that reciprocal ties are only within the two poles, and there is no such activity between the poles. The lack of reciprocal ties between poles reinforces the idea that this is a highly polarized event, in addition to the presence of echo chambers in each pole. There are no interactions between the users of each pole, and most users in this network prefer to interact with like-minded people. This reduces the probability of a user changing their views on the matter. It can be suggested that most of the users in this network, who have either tweeted or retweeted a hashtag relevant to the election, have participated in this discussion aware of their beliefs and with no intention of being influenced by other ideas. This is consistent behavior with users who join the discussion with the agenda to influence others. The communities of this network are labelled using the method mentioned before. The accuracy of the community labelling measured in this network is 100%, which suggests that the dark pink community in the graph is "pro voting," and the green community is users who are against the election and are "anti-voting." Although the anti-subgraph is much smaller than the Pro subgraph, the structure of the poles is very similar.

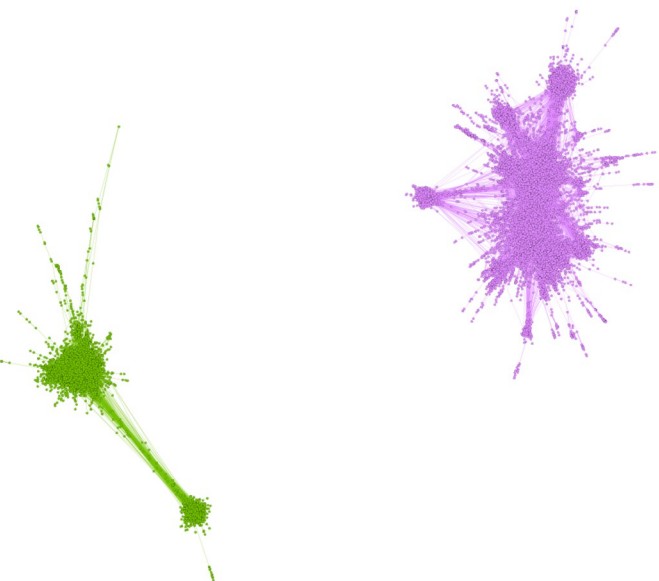

**Fig 5. The reciprocal subgraph of the retweet network of the election for the whole time of data gathering.**

Fig 6(a) and 6(b) show the radar graph for the anti and pro-community. It is demonstrated that in both these subgraphs; First, the assortativity coefficient is negative, suggesting that hubs are more likely to link to low degree nodes. Second, properties such as; diameter, average clustering coefficient, and modularity all deviate significantly from a random graph with the same number of nodes and edges. Lastly, the modularity of the pro-subgraph is much more than the modularity of the anti-subgraph, which indicates that there are more evident communities in the Pro subgraph.

In order to get a better sense of the users behind the tweets, The self-reported locations of the users in each pole is investigated. Fig 6(c) and 6(d) demonstrate the portion of users who declared to live in Iran and the ones who declared to live elsewhere in Anti and Pro community respectively. The unidentified group refer to users who either did not report their location, or did not use an actual place for their location (for instance some reported places like, "Hell" or "Hogwarts"). It is evident that users in the Pro community were more likely to declare Iran as their current residence.

To identify the type of users who were significantly influential in the network, Botometer API [30] is used. In this article, the top 1% of most retweeted users in each community in each period is regarded as the most influential users. The Cap score and the relative score for fake followers are extracted from Botometer API for all the most influential users. Cap score is a Complete Automated Probability of botscore that indicates what percentage of users with higher scores are bots. The score for fake followers is a relative number between 0 to 5, which specifies the number of fake followers each user has.

The users were again examined one week after the data gathering process (2 weeks after the election). It was discovered that 7.53% of the users in the Pro community and 7.16% of the users in the Anti community were either deleted or suspended. The deactivation of this many accounts so close to the trend suggests that such accounts only appeared to participate in the election discussion by either tweeting or retweeting a relevant hashtag which points to online manipulation. Moreover, the average cap score of the 1% most retweeted accounts in the pro

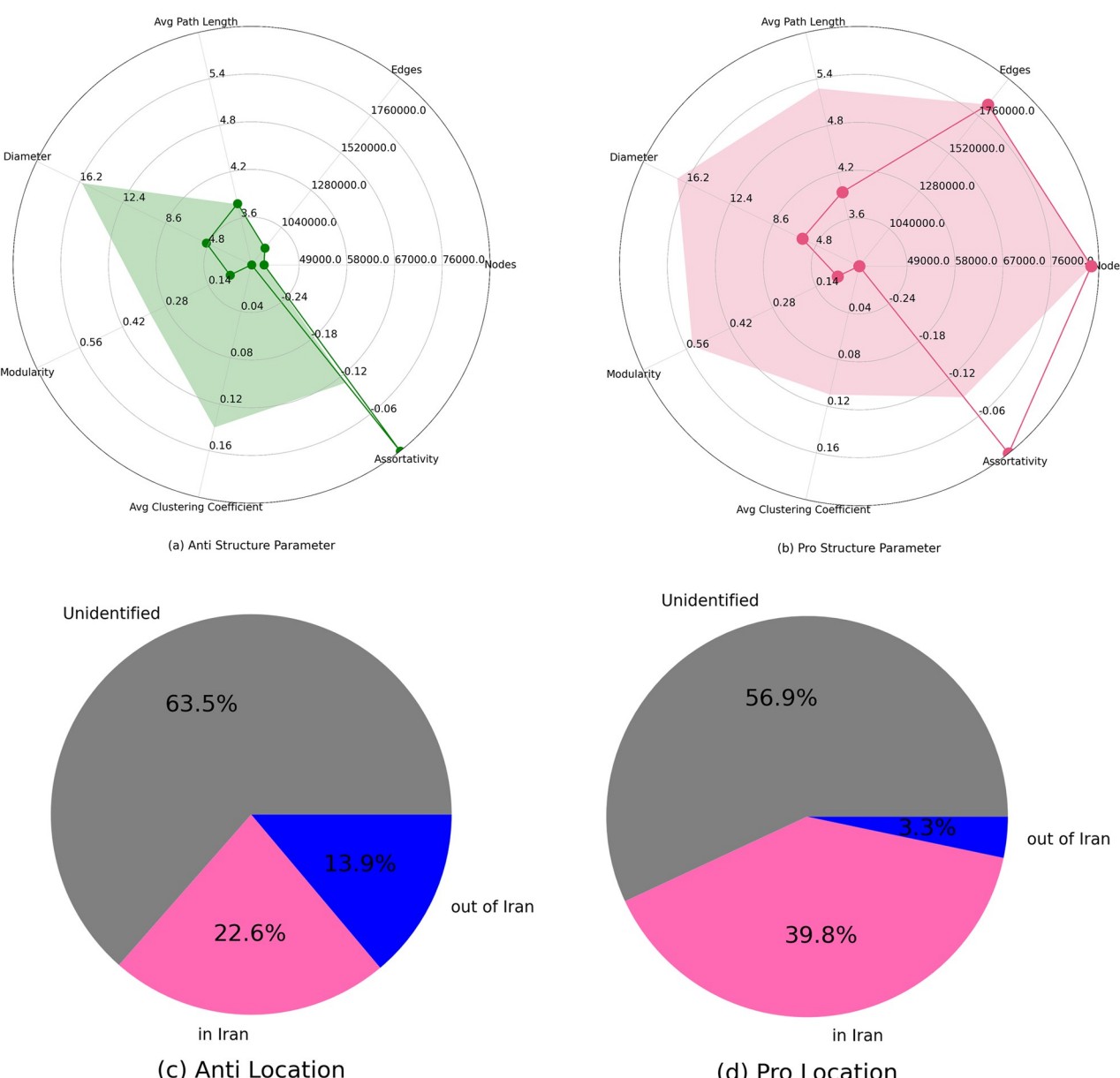

**Fig 6.** (above) radar graph of structural properties the two main subgraphs in the retweet network. The properties are; number of nodes, number of edges, diameter, average path length, modularity, assortativity, average clustering coefficient. The solid line represents the properties of a random graph with the same amount of nodes and edges. (a) Anti subgraph (b) Pro subgraph. (Below)pie graph of the proportion of the users in each community that reported to either live in Iran or elsewhere. (c) Anti subgraph (d) Pro subgraph.

and anti-communities are 0.81 and 0.78, respectively. This indicates that in both communities, hub accounts were more likely bots. It should be noted that there are no genuine users (accounts with a cap score of lower than 0.3) in the hubs of the pro community, which shows that the more influential accounts in this community all have automated or manipulatory behavior.

Fig 7 consists of violin plots that demonstrate the numerical distribution of Cap score and fake followers of the top 10% of the most retweeted users in users of each community. The

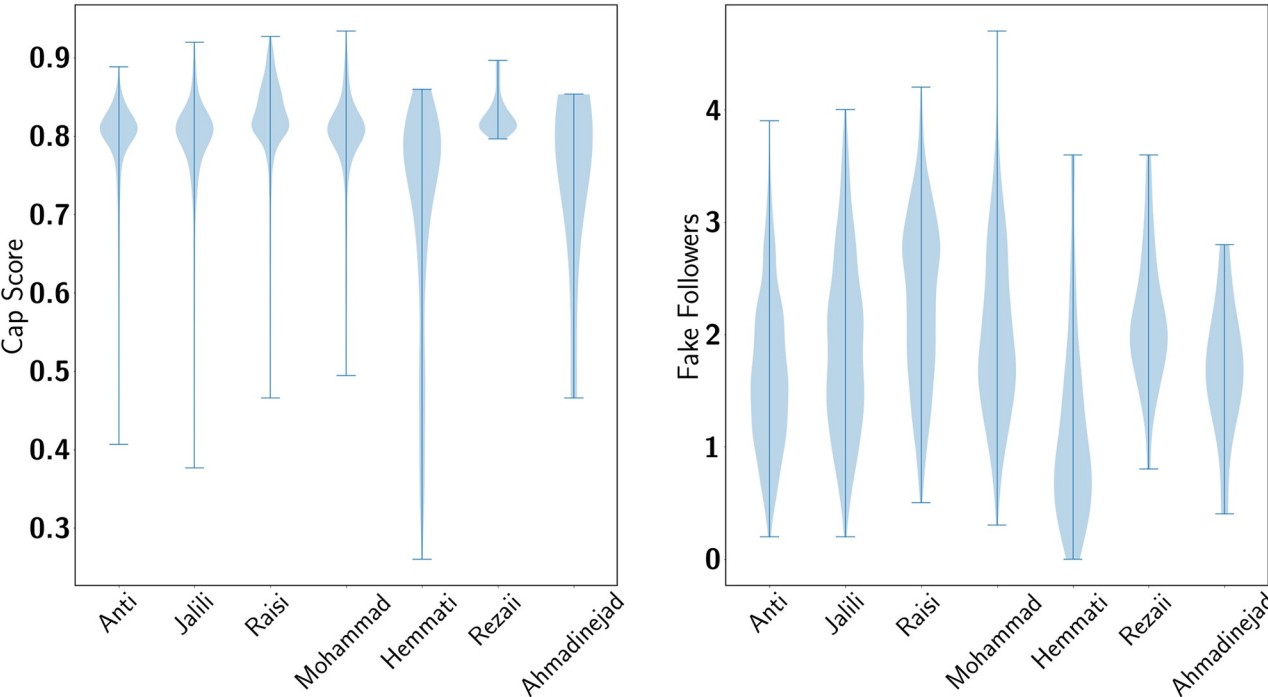

**Fig 7. (a) CAP (Complete Automation Probability) Score for the top 1 percent most tweeted users from each community (b) The relative score for fake followers for the top 1 percent most retweeted from each community.**

significantly high Cap score among all the communities stipulates the remarkable presence of bots among these users. It is essential to mention that these users are responsible for more than 90% of the edges within the network, accounting for most of the activities.

The presence of bots is highly significant among communities involved in both ends of the network, i.e., Anti and Raisi, Jalili, and Mohammad. However, two communities between the poles, Hemmati and Ahmadinejad's, have notably lower cap scores that point to more genuine users among their outspoken supporters. Their fake followers' relative score supports this claim and is noticeably lower than other communities. On the other hand, Rezaii's community was also mainly located between the two ends of the network; Fig 7 shows a concentrated distribution of high cap scores in the most influential users that can be construed as most of Rezaii's influential supporters are either campaign accounts or bots. The reason behind the high botscore of users with the most retweets might be because this type of user acquires other non-genuine users to retweet them to propagate their message [30, 44]; however, the mere presence of these accounts in some communities more than others points to organised activity to influence public opinion by these political groups. As it is mentioned before, a small number of nodes have more than 90%of the edges, creating what can be regarded as a core that is responsible for the network structure and the flow of information to a cloud that has a more significant number of users with much less activity where it can be assumed most of the genuine users exists. Further analysis suggests that "cores" in the opposite poles do not interact. However, they do interact within a pole in different communities. Considering the evidence of the high level of bot activity in all communities alongside other observations such as the way some communities were working together to gain support and leave the discussion in favour of the other at a sensitive point, all point to an organised campaign from all parties involved, to attempt to control the public opinion. In a non-political event, this number of bots and such

actions could lead us to an advertising campaign that wants to gain its audience's attention. However, this level of bot activity during an important political event, like an election, could only point to organisations or political groups coordinating actions to manipulate public opinion. Our results suggest that multiple political groups were organising such activities in this case.

## Conclusion

As the world evolves into being digitized, the impact of social media platforms on socio-political is getting more vital than ever before. In the Iran 2021 presidential election, there was a noticeable rise in political content, specifically Farsi twitter, whereas the physical events and advertisements were significantly reduced. While this paradigm shift has provided necessary space for free discussion in a pandemic era, it has also converted a lot of crucial elements of the physical world into different things.

The results indicate a high presence of bot activity among the most outspoken users of all communities during the 2021 election, suggesting organized activity. The retweet networks of the election preserve the same structure over the period leading to the election day: a polarized network with one pole representing the supporters of presidential candidates and the other pole for users promoting the boycott of the election. This arrangement is an irregular observation compared to other presidential elections, where the polarization is among candidates from opposing political parties. Even though the retweet network was not deformed as we got closer to the election day, we observed users' migration from different communities to the president-elect. At one point, the community that represented supporters of the presidential candidate, Saeed Jalili, merged into supporters of Ebrahim Raisi, president-elect, four days before Jalili officially suspended his election campaign. This could indicate that he opted out of the election due to his supporters' migration, or it could be a tactical move from his campaign. To summarize;

- Our study of the Iran 2021 election indicates that the Farsi Retweet network is an effective instrument to identify political debate, analyze and even predict political dynamics, and specific campaign strategies such as bots or automated accounts to influence public opinion.

- Unlike other elections, our studies reveal that the competition was not between the presidential candidates but between two large isolated poles, one encouraging people to vote and the other for voter suppression. Most of the candidates worked together in favor of the president-elect. This indicates that the opposition party did not have an acceptable candidate in the election, and most of the candidates in the race were from the right-wing (who were strongly supported by the state). This is a notable difference in this election compared to the previous Iranian election. By merely studying the retweet network of the election, one could recognize the structure of the competition within the election is telling of how the Persian Twittersphere is representative of the political system in Iran.

- The retweet network of the election in Farsi Twitter is highly polarized, with an E-I index of -0.98. The reciprocal ties as a sign of social communication are only present within the poles and in the communities inside the poles. This finding confirms the existence of two echo chambers, one in favor of the election and the other to boycott the election.

- There are pieces of evidence that Iranian politicians in all groups have noticed the influence of social media on public opinion and have invested in orchestrating online movements in their favor.

- All the invested parties have used campaign accounts or bots to propagate their agenda in this election. Since they are the most retweeted accounts in all communities, these accounts have an average score of 0.85, above the threshold for non-genuine users.

## Supporting information

**S1 Appendix. Trending hashtags.**
(PDF)

## Acknowledgments

We would like to thank Prof. Shant Shahbazian, Prof. Afshin Montakhab, Prof H. Reza Sepangi and Prof. Behzad Ghanbarian for their guidance and constructive comments regarding this manuscript.

## Author Contributions

**Conceptualization:** Saeedeh Mohammadi, Parham Moradi, S. Mahdi Firouzabadi, Gholam-reza Jafari.

**Data curation:** Saeedeh Mohammadi, Parham Moradi.

**Formal analysis:** Saeedeh Mohammadi, Parham Moradi, Gholamreza Jafari.

**Investigation:** Saeedeh Mohammadi.

**Methodology:** Saeedeh Mohammadi, Gholamreza Jafari.

**Project administration:** Gholamreza Jafari.

**Visualization:** Saeedeh Mohammadi.

**Writing – original draft:** Saeedeh Mohammadi.

**Writing – review & editing:** Saeedeh Mohammadi, Parham Moradi, S. Mahdi Firouzabadi.

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
