## [Decision Letter · Decision Letter 0]

12 Apr 2022

PONE-D-22-05615The Footprint of Campaign Strategies in Farsi Twitter: A case for 2021 Iranian presidential electionPLOS ONE

Dear Dr. Jafari,

Thank you for submitting your manuscript to PLOS ONE. After careful consideration, we feel that it has merit but does not fully meet PLOS ONE’s publication criteria as it currently stands. Therefore, we invite you to submit a revised version of the manuscript that addresses the points raised during the review process.

We look forward to receiving your revised manuscript.

Kind regards,

Gábor Vattay, PhD, DSc

Academic Editor

PLOS ONE

Journal Requirements:

Reviewers' comments:

Reviewer's Responses to Questions

**Comments to the Author**

1. Is the manuscript technically sound, and do the data support the conclusions?

Reviewer #1: Partly

Reviewer #2: Yes

2. Has the statistical analysis been performed appropriately and rigorously? 

Reviewer #1: Yes

Reviewer #2: Yes

3. Have the authors made all data underlying the findings in their manuscript fully available?

Reviewer #1: Yes

Reviewer #2: No

4. Is the manuscript presented in an intelligible fashion and written in standard English?

Reviewer #1: Yes

Reviewer #2: No

5. Review Comments to the Author

Reviewer #1: Comments on: The Footprint of Campaign Strategies in Farsi Twitter: A case for 2021 Iranian presidential election

This paper aims to investigate coordinated and organized activities in Persian Twitter during the 2021 presidential election. The manuscript is well structured and written. Also, I agree with authors that we need more work on Iranian Twittersphere to better understand how and to what extent organized activities are shaped and operated in this space. Though, I believe that this work has certain potentials to contribute to our knowledge of Twitter activism, by bots and by genuine users, there are some major and minor revisions that should be addressed first. I outline the most important of these revisions below.

- The manuscript argues that it intends to analyze organized activities on Persian Twitter, but it did not do so as the data and analyses did not provide sufficient and substantial evidence of such activities. It is the major shortcoming of the current work. In fact, findings are quite descriptive. The findings section mainly provides a descriptive account of the rise and fall of communities in Persian Twitter. Moreover, it indicates that most of these communities were populated by bots. While these findings are interesting, authors need to go deeper to investigate more substantial questions and shed light on more complicated dynamics of bots’ activism on Persian Twitter. Moreover, it is not enough to simply state that Twitter communities were home of bots and automated accounts. You need to give more details, e.g., how much of nodes in each cluster were bot? which factors affect the number of bots in each community? What is the ratio of bots to genuine users in each cluster? Then you should discuss what these numbers mean in terms of organized activities.

Authors also mentioned time to time that some changes in the network, such as disappearing a community or merging communities with each other, are the evidence of organized activities on Persian Twitter (for instance; lines 211-213). However, they did not provide enough information to support their arguments. It is not enough to just mention that there are some coordinated actions based on the number of bots or the changes of communities. You should go deeper to explore how these bots worked and were constituted. The manuscript should provide more details on how organized activities on Persian Twitter were operated and to which extent? For instance, how and by what candidates/parties were bots employed? To which extent? Were there any similarities/differences between bots activism in different communities? How did automated accounts interact with other users to manipulate the flow of content? How did they try to intervene with genuine users’ practices in shaping counter narratives and discourses? Instead of dealing with such questions that can enhance our understanding of organized activities on Persian Twitter, the manuscript just gave a descriptive explanation of the changes in communities on Twitter.

A related point; since the findings are descriptive, authors were unable to provide some solid and explanative conclusions. The conclusion section raises several points which were not supported by data and analyses. I provide some examples below:

o In the first paragraph, you suddenly mentioned the Covid-19 pandemic. Since it is not the focus of your research, and you did not discuss it before, I suggest consider removing it.

o In this section, you made some points about the anonymity of users and raised some abrupt arguments about the anonymity of users on Persian Twitter. First, I can’t see how it is relevant to your research. Furthermore, the problem of anonymity of users on Persian Twitter is more complicated than can be concluded in a few lines. I suggest to base your conclusions on your data and analyses. None of these conclusions were supported by your work.

o The manuscript mentioned in lines 296-298: our current study of the debate network of the Iran 2021 election reveals that the main competition was not between the presidential candidates but between two large isolated poles, one encouraging people to vote and the other for voter suppression. I am wondering why authors did not consider that this could be pertinent to the fact that most of candidates were deprived from participating in election by the Guardian Council. In fact, the competition was not between candidates since most of them were belonged to a same political camp which was supported by the state power.

o You argued in line 285-287: Our study of the Iran 2021 election indicates that the Farsi Retweet network is an effective instrument to identify political debate, analyze and even predict political dynamics and campaign strategies. Your research did not carry out these jobs. For instance, how did your work analyze campaign strategies?

Other conclusions also should be supported by your data.

Other points:

- Authors should provide more context on politics and Twitter in Iran. They discussed several candidates’ communities in findings section, but these candidates are unknown to the readers who are not familiar with Iran’s political landscape. Furthermore, they should discuss the political atmosphere in which 2021 election was held. For instance, the disapproval of many reformist candidates which made many citizens angry. Or the nationwide protests in November 2019 which affected the rate of participation and the discussions on social media significantly.

- I can’t agree more with authors in this fact that there is an academic gap in analyzing coordinated activities on Persian Twitter. But I believe there are many studies in this topic on other countries. Authors should review the existing literature and link this work to them. In this way, they can show what is the added value of the current study. Doing so, they will also able to highlight the contribution of this research to this field beyond Persian Twitter.

- Authors need to compare and contrast the findings with those analyzing previous presidential elections, such as Kermani and Adham (2021) work on 2017 election, and Khazraee (2019) work on 2013 election. They should discuss how the community of users has been transformed during the time to highlight the novelty of their findings.

- Authors provide comprehensive details about the methodology. However, I suggest to remove some parts of this section if they could be found in reference books. If they devised a new approach or measurement, then it should be clearly mentioned. Instead, I suggest to discuss what these measurements mean and what they did in this research.

- I also suggest to get involved more critically with literature into social media activism and coordinated action to disrupt it.

Some minor points:

- I guess the columns of No. users, and No. tweets should be exchanged in Table 1. They seem wrong now.

- MEK is a militia group. Authors need to provide more information on this group, and also mention what this abbreviation stands for.

- Several sentences need proper references throughout the paper.

- I don’t understand this sentence: Note that this number differs from node’s in-degree since the network is constructed from only the tweets that include a trending hashtag about the election (85-86).

This manuscript has some promises and I do like to see it is out. I wish the authors success in revising the paper to turn it to a more solid work. I think the paper could contribute to Twitter studies in Iran and fill some existing gaps. I know how valuable doing these researches in Iran are, and I do appreciate the authors endeavor in conducting such an innovative and thoughtful study in Iranian context. Finally, I hope the authors find the comments helpful and wish them luck in their academic career.

Best wishes.

Reviewer #2: This paper is scientifically sound, but the presentation is very very poor.

The introduction should be at least three paragraph: motivation, the paper’s contribution in the larger literature, and the data and results of the paper. Much of this information is in the one introductory paragraph and just needs to be split.

I have questions about the data collection. First, I would like to see a table, in the appendix is fine, of the 97 hashtags. Second, I need to know more. Is it Twitter that identified if they are trending? Is 97 all of the trending hashtags during the period? Third, did the authors connect to the streaming API, query a search endpoint in real time, or use the Academic Research product after the fact? Fourth, why divide the period into weeks instead of aggregating by days? There are enough users and tweets to analyze the data by day instead of by week.

The authors claim retweets are discussions. I disagree. A quote retweet could be a discussion, but a retweet just means engagement. The authors should be clearer on this point. Moreover, if they want actual discussion, they could use the conversation ID that Twitter now gives, in v2 of the API, to every tweet and its replies.

Next, tell me something about the location of who is tweeting; the authors only do so for MEK and then only obliquely. Though Twitter is banned in Iran, there is evidence that many people there use it; see this blog post by Layla Hashemi and Steven Wilson that does a good job of identifying users in Iran: https://www.washingtonpost.com/politics/2020/01/08/twitter-is-where-iranian-dissidents-might-support-soleimanis-killing-opposite-happened/. Two other ideas for location: VPNs do not change users’ location, so some of the tweets may actually show up as being from Iran (this is what Roberts, Hobbs, and Steinert-Threlkeld do for their work on China), and look at users’ self-reported profile location. You could also look at the timezone of the user profile. At the end of the day, the results are not wrong or right if they are driven by individuals outside of Iran, but knowing that factoid will make the results clearer.

My biggest critique is that the figures are so blurry they are essentially unreadable. I have to believe they would not pass typesetting review, and if they appeared in PLoS One in their current form it would be an embarrassment to the journal and authors. The blurriness is not a hardware issue, as the authors clearly have enough computing power to analyze a large quantity of data and generate visuals of them. They need to be much sharper and with larger font. Otherwise, the reader cannot interpret the results and therefore is left trusting the authors.

Finally, there are very many typos that I fear reflect a lack of attention to detail, not English as a second language. For example, see:

• “, Where”

• “Eventually, The”

• “Hashtag” needs to be lowercase unless it starts a sentence.

• ‘’Twitter Machine” has close quotes at the start, fix.

While there are definitely syntactical issues that arise since the authors are not native English speakers, those are largely fine and understandable. The issues is that there are very very many instances of inconsistent syntax that suggest not lack of understanding of English but lack of careful proofreading.

6. PLOS authors have the option to publish the peer review history of their article (what does this mean?). If published, this will include your full peer review and any attached files.

Reviewer #1: **Yes: **Hossein Kermani

Reviewer #2: No

---

## [Author Response · Author response to Decision Letter 0]

26 May 2022

We thank the reviewers for their critical assessment of our work as their constructive criticism has improved the work immensely. 

Since the response included a figure and a table, it is uploaded in a PDF by the name "Response to Reviewers". In which we address their concerns point by point. The authors detailed answers are written in blue, and the pieces added to the article are written in color red.

---

## [Decision Letter · Decision Letter 1]

9 Jun 2022

PONE-D-22-05615R1The Footprint of Campaign Strategies in Farsi Twitter: A case for 2021 Iranian presidential electionPLOS ONE

Dear Dr. Jafari,

Thank you for submitting your manuscript to PLOS ONE. After careful consideration, we feel that it has merit but does not fully meet PLOS ONE’s publication criteria as it currently stands. Therefore, we invite you to submit a revised version of the manuscript that addresses the points raised during the review process.

In the light of the reports, I see that both reviewers are now satistfied with mansuscript. One of the reviewers thinks that more could be done to address some important issues. Please, consider these comments and address the issues raised by the reviewer. I think, some of them can be addressed with some minor research effort, some can be postponed to 'future work to be done' in the concluding part of the manuscript.

We look forward to receiving your revised manuscript.

Kind regards,

Gábor Vattay, PhD, DSc

Academic Editor

PLOS ONE

Journal Requirements:

Reviewers' comments:

Reviewer's Responses to Questions

**Comments to the Author**

1. If the authors have adequately addressed your comments raised in a previous round of review and you feel that this manuscript is now acceptable for publication, you may indicate that here to bypass the “Comments to the Author” section, enter your conflict of interest statement in the “Confidential to Editor” section, and submit your "Accept" recommendation.

Reviewer #1: (No Response)

Reviewer #2: All comments have been addressed

2. Is the manuscript technically sound, and do the data support the conclusions?

Reviewer #1: Yes

Reviewer #2: Yes

3. Has the statistical analysis been performed appropriately and rigorously? 

Reviewer #1: Yes

Reviewer #2: Yes

4. Have the authors made all data underlying the findings in their manuscript fully available?

Reviewer #1: Yes

Reviewer #2: Yes

5. Is the manuscript presented in an intelligible fashion and written in standard English?

Reviewer #1: Yes

Reviewer #2: Yes

6. Review Comments to the Author

Reviewer #1: I appreciate the authors' effort in revising the manuscript. It is now read better, and the contribution is more evident. Also, the revised parts add to the clarity of the manuscript. However, two major points are still unanswered. First, as I mentioned in the previous round, the focus of this work is currently on the RT network in Persian Twitter during the 2021 election, not the coordinated activities. The findings provide a descriptive account of the RT network and fail to go deeper to investigate the coordinated activities on Persian Twitter. Since the latter is the aim of this work, the focus should be on examining the bot and coordinated actions with more details. The authors mentioned in their letter that they found many deleted or suspended accounts. These findings are of significant importance in analyzing coordinated actions. They could detail these findings, discussing how many of the accounts in each community are bots or automated, as well as the portion of the accounts that had been suspended or deleted. Of course, I do not mean they do that on the whole network. They could focus on a particular portion in each cluster. Also, the authors mentioned that the number of bots in all communities is the same. In this step, I suggest they should examine the automated accounts' communicative practices and strategies. They can compare these mechanisms within pro-regime clusters and between pro and anti-regime communities. Such analyses add to the dept of the research and enable authors to provide more substantial statements and conclusions.

Also, we know already that MEK and monarchist groups use bots on a large scale on Twitter. Researchers could try to examine if there are other groups using bots in anti-community or not? In each case, they can extend their analyses by comparing the results with the bot activism in the pro-regime cluster. As another suggestion, the authors mentioned that there are no reciprocal ties between poles. There are mainly in-group relations. Elaborate more on such findings instead of discussing the changes in communities!

Moreover, while they mentioned some works on Persian Twitter and coordinated activities in other countries, they did not compare and contrast their result with the existing literature. They should go further than simply mentioning some previous studies. They should clearly show the links between their work with the existing body of research to highlight the contribution of this study.

I am generally in favor of publishing this work. There are promises in the manuscript. Also, the revised version is a significant improvement. If the authors could address the above-mentioned concerns, I guess this manuscript would contribute to Persian Twitter studies as well as the existing literature on coordinated activities on Twitter significantly.

Reviewer #2: Good job responding to the comments. I really do not have anything to say, I am typing because PLoS ONE requires a 100 character response for me to submit the review.

7. PLOS authors have the option to publish the peer review history of their article (what does this mean?). If published, this will include your full peer review and any attached files.

Reviewer #1: **Yes: **Hossein Kermani

Reviewer #2: No

---

## [Author Response · Author response to Decision Letter 1]

20 Jun 2022

We thank the reviewer for his constructive suggestions. The response to the points made by the reviewer has been uploaded.

---

## [Editor Report · Decision Letter 2]

21 Jun 2022

The Footprint of Campaign Strategies in Farsi Twitter: A case for 2021 Iranian presidential election

PONE-D-22-05615R2

Dear Dr. Jafari,

We’re pleased to inform you that your manuscript has been judged scientifically suitable for publication and will be formally accepted for publication once it meets all outstanding technical requirements.

Kind regards,

Gábor Vattay, PhD, DSc

Academic Editor

PLOS ONE
---

## [Editor Report · Acceptance letter]

27 Jun 2022

PONE-D-22-05615R2 

The Footprint of Campaign Strategies in Farsi Twitter: A case for 2021 Iranian presidential election 

Dear Dr. Jafari:

I'm pleased to inform you that your manuscript has been deemed suitable for publication in PLOS ONE. Congratulations! Your manuscript is now with our production department. 

Kind regards, 

on behalf of

Dr. Gábor Vattay 

Academic Editor

PLOS ONE